# Degradation of Sub-Micrometer Sensitive Polymer Layers of Acoustic Sensors Exposed to Chlorpyrifos Water-Solution

**DOI:** 10.3390/s22031203

**Published:** 2022-02-05

**Authors:** David Rabus, Fanny Lotthammer, Joscelyn Degret, Lilia Arapan, Frank Palmino, Jean-Michel Friedt, Frédéric Cherioux

**Affiliations:** Franche-Comté Électronique Mécanique Thermique et Optique-Sciences et Technologies, Université de Franche-Comté, CNRS, 15B Avenue des Montboucons, CEDEX, F-25030 Besancon, France; david.rabus@femto-st.fr (D.R.); fanny.lotthammer@femto-st.fr (F.L.); joscelyn.degret@edu.univ-fcomte.fr (J.D.); lilia.arapan@femto-st.fr (L.A.); frank.palmino@univ-fcomte.fr (F.P.)

**Keywords:** SAW, pesticide, sensing layer, aqueous stability

## Abstract

The detection of organophosphates, a wide class of pesticides, in water-solution has a huge impact in environmental monitoring. Acoustic transducers are used to design passive wireless sensors for the direct detection of pesticides in water-solution by using tailored polymers as sensitive layers. We demonstrate by combining analytical chemistry tools that organophosphate molecules strongly alter polymer layers widely used in acoustic sensors in the presence of water. This chemical degradation can limit the use of these polymers in detection of organophosphates in water-solution.

## 1. Introduction

Monitoring the increasing number of pollutants in the water table is an ongoing concern for water treatment systems and water resource managers [1]. Pesticides, heavy metals, personal care products, natural toxins, and a host of other organic and inorganic chemical pollutants and their products can increase toxicity in water. Among all in the class of water pollutants, pesticides based on organo-phosphates are one of the main hazardous compounds due to their long-term toxicity. The development of innovative sensors that can identify the presence of chemical pollutants in various types of water without preparation for continuous monitoring, as opposed to sampling as needed for current laboratory methods, is still challenging [2,3]. In that sense, direct detection sensors relying on the measurement of a physical quantity upon reaction of an analyte, such as an acoustic velocity in surface acoustic wave (SAW) sensors, have been developed [4,5,6,7,8,9,10,11,12]. However, in all cases, the sensing physical principle is not selective, and the sensor selectivity is brought by the chemical functionalization of the surface. Polymers offer a variety of sensing layers and can be directly integrated in collective processes of fabrication [13,14]. For instance, polymers such as polyepichlorohydrin (PECH), polyisobutene (PIB), and poly (butyl methacrylate) (PBMA) have been widely used for the development of chemical SAW sensors [15,16,17]. However, polymers also being chemical compounds, they could react with pollutants, which could affect their stability and finally could modify their sensing ability: despite having been used for BTEX detection, using such sensors in water tables contaminated with pesticides might alter their detection capability as will be discussed here. We investigate the stability of these polymers widely used as sensitive layers for the direct detection of an organophosphate pesticide in water-solution, which is representative of the environmental conditions. Within this objective, we choose chlorpyrifos as model of organophosphate pesticide [18]. The detection of chlorpyrifos has already been investigated in gas [19,20] or aqueous [21] phase by using SAW sensors under controlled conditions.

In this work, we monitor the acoustic response of SAW devices functionalized with PECH, PIB and PBMA thin films exposed to chlorpyrifos water-solution in environmental conditions. Then, we assess the instability of the measurements by considering analytical characterization of the thin films prior and after exposure to chlorpyrifos using Fourier-Transform InfraRed Spectroscopy (FT-IR) and Atomic Force Microscopy (AFM). We demonstrate the irreversible degradation of the thin polymer films exposed to chlorpyrifos in water-solution. This instability of sensitive layers seems to be a strong limitation to the popularization of this kind of acoustic sensors in environmental challenge and water resource management in practical deployment scenarios.

## 2. Materials and Methods

### 2.1. Design of Sensors and Active Sensitive Layers

#### 2.1.1. Design of Sensors

Surface Acoustic Wave (SAW) reflective delay lines were manufactured by depositing 200 nm thick aluminum interdigitated electrodes acting as transducer (IDT in Figure 1) and mirrors (M1-4 in Figure 1) on YXl/36° lithium tantalate substrate. The split finger transducer and mirrors generate a shear leaky SAW confined to the surface by metallizing the areas between IDTs with a wavelength of 41 μm with a metallization ratio of 50% and an acoustic aperture of 1600 μm, leading to a center frequency around 100 MHz. Both the IDT and mirrors are apodized, with the mirrors designed for reemission of the incoming wave rather than mechanical reflection: mirrors M1-4 act in a regenerative mode where the incoming wave polarizes each mirror, regenerating a SAW towards the transducer (IDT in Figure 1). The four mirrors were included to define four areas free of transducers acting as sensing areas coated with the same metal to confine the pseudo-shear wave to the surface and coated with polymer for sensing capability. The distance from IDT to M1 is 1700 μm, from IDT to M2 is 3400 μm, from IDT to M3 is 4700 μm, and from IDT to M4 is 6200 μm, leading to echoes typically delayed by about 800 ns from each other depending on the acoustic velocity, which itself depends on the polymer layer thickness, starting at 4200 m/s when no polymer is present.

The four areas are selected with two areas patterned with the same sensing polymer to cancel the impact of the probing electronic unit to sensor range (electromagnetic time of flight cancellation by differential measurement); a third area patterned with a polymer insensitive to the compound to be detected for temperature compensation; and a fourth area used here to assess reproducibility, but which could be used to pattern a second sensing polymer to improve selectivity by inducing a different response to the same compound.

Thin polymer films are spin-coated to reach thicknesses between 500 and 1200 nm aimed at confining acoustic energy in a Love mode guided in the polymer (see below). The IDT is wired to an Agilent E5071B network analyzer for probing the sensor acoustic response once every 30 s by recording the S_11_ transfer function on a 20 MHz bandwidth—a 20% relative bandwidth selected as greater than the sensor transfer function width determined by the electromechanical coupling coefficient—in order to post-process the spectral domain response to reach the time domain response by inverse Fourier-transform. Echoes are identified as local maxima of the magnitude, and the phase is recorded at the associated time delay found, by design of the reflective SAW delay line, between 0.8 and 2 microseconds (Figure 2).

#### 2.1.2. Active Sensitive Layers

All commercially available starting materials and solvents were purchased from SigmaAldrich, Fisher and used without further purification. All polymer deposits were performed in a Class-100 clean-room on the Karl Suss RC-8 CT-62 spin-coater. All surface activations were performed on PVA/Tepla Giga BATCH 360 M. The thickness of polymer layers was performed with a stylus profilometer Dektak xt (C9). All layers of polymers were characterized by Fourier-transform Infrared-Spectroscopy (FT-IR) by using a spectrum two FT-IR spectrometer from Perkin Elmer. AFM measurements were performed with a Bruker Icon AFM connected to a Nanoscope V controller. The AFM was employed in peak-force tapping mode using Scanasyst-Air-HPI Bruker probes of nominal stiffness of 0.4 N m^−1^ and a nominal tip radius of 2 nm.

### 2.2. Deposition of Polymers

#### 2.2.1. Polymers Films for SAW Devices

PBMA (90 mg) was dissolved in toluene (1000 mg) and stirred for 30 min at room temperature. The chip sensor surface (aluminum and lithium tantalate) was activated by plasma (O_2_, 300 W, 10 min). A volume of 250 µL of polymer solution was then deposited on clean surface by using a plastic pipette. PBMA solution was spread at a speed of 2500 rpm with an acceleration of 800 rpm/sec for 30 s. The polymer layers covering contact pads were removed with a cotton-bud dipped in acetone. Then, an annealing at 100 °C for 2 min give a clear, translucent, and homogenous polymer film with a thickness of 980 nm.

#### 2.2.2. PBMA Films for FT-IR Analysis

PBMA (42 mg) was dissolved in toluene (1000 mg) and stirred for 30 min at room temperature. Silicon wafers (size: 3′′, thickness: 380 µm) were cut into 1.5 cm × 2 cm pieces. Then, they were cleaned in piranha solution for 2 min and rinsed with clean water. Silicon surface was activated by plasma (O_2_, 300 W, 10 min). A volume of 250 µL of polymer solution was then deposited on clean surface by using a plastic pipette. PBMA solution was spread at a speed of 2500 rpm with an acceleration of 800 rpm/sec for 30 s. Then, an annealing at 100 °C for 2 min give a clear, translucent, and homogenous polymer film with a thickness of 200 nm. We used a thinner silicon substrate and a thinner layer of polymer for FT-IR analysis than for AFM or SAW devices in order to decrease the light absorption.

#### 2.2.3. PBMA Films for AFM Analysis

A similar procedure used for AFM sample preparation was then used for FT-IR analysis. Polymer solutions were deposited on silicon wafer parts (1000 µm, 1.5 cm × 1.5 cm). The thickness of resulting polymer layers was 500 nm.

#### 2.2.4. Thickness Measurements

Dektak Thickness measurements were collected by using this set of parameters, with range of 6.5 µm, hills and valleys as profile, 0.5 mg for stylus force, and a resolution close to 0.1 µm/pt.

Thin scratches were applied to the corner and to the center of the PBMA-coated pieces.

#### 2.2.5. Solution of Chlorpyrifos

An amount of 40 mg of chlorpyrifos was dissolved into 40 mL of acetonitrile. The resulting solution was then diluted in 960 mL of pure water.

## 3. Results

### 3.1. Investigation of Sensing Capabilities

The sensor was designed for wireless sensing of an analyte in liquid phase: such conditions were simulated in a controlled laboratory environment by flowing solutions in an open-well liquid cell with fluids continuously injected and sucked by a peristaltic pump (Peripro-4HS from WPI) running at 5 mL/min. In order to obtain a homogenous aqueous phase, we used a water solution containing 4% of acetonitrile which strongly promotes the dissolution of chlorpyrifos. This solution is named ‘blank’ in the following parts. All measurements were simultaneously recorded on three parallel sensors for reproducibility assessment, and each cycle of blank-analyte-blank was repeated to assess the stability of the polymer layer. Phase differences between second and first echo on one hand, and between third and first echo on the other hand, aim at cancelling correlated noise such as fluctuations in the level of liquid in the flow cell. Each cycle was completed when an asymptotic behavior was reached, with kinetics lasting multiple hours as shown in Figure 3. Each exposure to chlorpyrifos exhibits a phase decrease indictive of mass loading usually associated with adsorption or absorption of a molecule in or on the polymer layer. The reaction is observed to be reversible since phase rises upon exposure to the blank.

We observe that the phase variations, decreasing being usually associated with mass loading when absorbing analyte molecules in the polymer matrix and increasing being associated with layer stiffening or desorption, are not consistent with injection steps of the pesticide or blank solutions. Indeed, the phase starts dropping upon exposure to the pesticide prior to rising again after 10 h exposure (Figure 3a). On the other hand, exposure to the blank solution around 20 h does not induce a stabilization of the signal or a rise that would be associated with desorption. The same effect is observed on all three parallel channels (Figure 3a and Figure 4a) and consistently by differential analysis of echo 1 with echo 2 (Figure 3b) or echo 1 with echo 3 (Figure 4b), eliminating the risk of experimental setup instability.

Magnitude of the insertion losses of echo 2 is shown in Figure 4a to emphasize the degradation of the polymer as visible with the increasing losses after each exposure and cleaning steps. The insertion loss increases of several dB are much larger than those usually observed upon polymer, globular protein, or DNA adsorption, hinting at the in-depth degradation process scattering the acoustic wave [22,23,24].

The same evolution for phase and amplitude is observed for PECH and PIB polymers (see Appendix A).

The lack of reproducibility on the quantitative evolution of the phase from one cycle to another leads to the investigation on the stability of the various polymers upon exposure to chlorpyrifos using analytical chemistry techniques.

### 3.2. Physico-Chemical Analysis of Sensitive Layers

In order to understand the variation and the non-reproducible phase variation of SAW devices, we investigated the morphology of the sensitive polymer layers by AFM. The three polymers were deposited by spin-coating onto a silicon substrate as a model of atomically flat surface. We performed three different AFM mappings on (i) the raw polymer layer after deposition by spin-coating onto silicon substrate; (ii) this polymer layer after an immersion into a water-acetonitrile (96:4) of chloropyrifos (40 mg/L) during 12 h; and finally (iii) the latter polymer layer after an immersion in water-acetonitrile (96:4) during 12 h. AFM was used to acquire images in the PeakForce Tapping mode of Bruker Icon. All images were recorded under ambient (air pressure, 298 K) conditions with 512 points × 512 lines.

The layers obtained by spin-coating a solution of PBMA, PECH or PIB onto a silicon substrate are flat with a few nanometers peak to peak roughness and homogenous, as shown by AFM images depicted in Figure 5, Appendix A, respectively.

In the case of PBMA layer, the roughness is below 2 nm for a length of 30 microns (RMS roughness: 0.3 nm, Figure 5a). This polymer layer was immersed for 12 h in a water-acetonitrile (96:4). The surface is rougher than before the immersion, with a peak-to-valley depth of 30–40 nm (RMS roughness: 11.3 nm, Figure 5b). After immersion in a water-acetonitrile solution of chlorpyrifos for 12 h, the polymer layer was rinsed for 1 h with an acetonitrile-water solution (96:4). The overall surface is significantly different with respect to the previous observations prior to rinsing: holes are observed on the surface, with depths close to 500 nm, equal to the thickness of the polymer layer, and a width varying from 1 to 10 microns (RMS roughness: 321.0 nm, Figure 5c).

The same characterizations were performed on PIB and PECH layers. AFM topography scans and height profiles show the same appearance of crossing holes after the rinsing with water-acetonitrile solution while the layers are only slightly affected with the three first steps (Appendix A in ESI).

### 3.3. FT-IR Spectra

Each polymer layers deposited onto a silicon substrate were investigated by FT-IR at different steps of the previous procedure described for AFM investigations.

The spectrum of PBMA layer (blue line in Figure 6) shows three strong bands centered at 2960, 2874 and 1729 cm^−1^, respectively. In the spectra recorded after exposition to water solution of chlorpyrifos (red line) and after rinsing with water-acetonitrile solution (yellow line), the three bands are still observed, and some new bands centered at 1576, 1542, 1436, 1412, 1023 and 846 cm^−1^, respectively, are clearly identified (Figure 6b).

In the case of PIB and PECH layers, similar behaviors are observed. New extra-bands are observed in FT-IR spectra, centered at 1164 and 1025 cm^−1^ and at 1022 and 967 cm^−1^ for PIB and PECH respectively, after immersion to chlorpyrifos water solution and rinsing with a water-acetonitrile solution (Appendix A in ESI).

## 4. Discussion

The phase and amplitude variations of the acoustic sensors upon exposure to a water-acetonitrile solution of chlorpyrifos are not reproductible (See Figure 3 and Figure 4). AFM experiments demonstrate that chlorpyrifos is involved in this irreproducibility of acoustic response (Figure 3 and Figure 4). Strong fluctuations which cannot be explained by the reversible adsorption of chlorpyrifos on the surface of PBMA, PIB or PECH polymers are observed. This effect is striking when comparing our curves to those obtained in the gas phase, which exhibit excellent reproducibility [20]. In addition, other work has been carried out in an aqueous medium but with other polymers and under other conditions [21]. Indeed, these authors used buffer solutions (with phosphate buffer solution (PBS)) at a fixed pH of 6.20 while this work uses an aqueous solution without PBS, but whose pH is varying from 6 to 7, depending on the experimental conditions, representative of water table monitoring conditions. In order to confirm the impact of the buffer in the polymer degradation prevention, we investigated by AFM the evolution of a layer of PBMA exposed to 100 mL of a PBS buffered water solution at pH = 6.2 containing chlorpyrifos (Figure 7).

In this case, no hole is observed. The layer before and after exposure to the buffered solution exhibit a similar roughness, on the order of a few nanometers as opposed to the holes penetrating through the polymer layer in the absence of PBS (Figure 5).

Consequently, we suggest that the degradation of polymer layers could be attributed to local variation of pH due to the presence of chlorpyrifos. Indeed, in aqueous media, chlorpyrifos is hydrolyzed which leads to the formation of 3,4,6-trichloro-2-pyridinol and O,O-diethyl hydrogen thiophosphate. These two molecules are weak acids, which can lead to the formation of H^+^ in water (Figure 8). As the chlorpyrifos is hydrophobic, this compound is poorly soluble in the water-acetonitrile (96:4) solution. Therefore, chlorpyrifos can be mainly adsorbed onto the PBMA thin layer, because PBMA contains alkyl chains promoting van der Waals interaction with the aromatic core of chlorpyrifos. Finally, as chlorpyrifos is mainly located at the water–polymer interface, the formation of H^+^, due to the hydrolysis of chlorpyrifos, is enhanced at this interface, which leads to the enhancement of acidic degradation of the PBMA layer as observed in the AFM images. In addition, pH can be very different at solid-liquid interface from the solution as it has been reported in the case of polymer–water interface [25]. Figure 9 emphasizes the role of acetonitrile in dissolving the products coming from the acidic hydrolysis of PBMA polymer strands. Indeed, exposition of the PBMA layer to the same pH without acetonitrile leads to surface degradation but shallower holes (a peak-to-valley depth of 20 nm with RMS roughness of 4.8 nm, Figure 9a; a peak-to-valley depth of 40 nm with RMS roughness of 10.3 nm, Figure 9b). Similarly, Appendix A confirms this conclusion by replacing constant pH solution with constant chlorphyrifos concentration and varying acetonitrile concentration: halving the latter significantly reduces the degraded polymer solubilization capability and hence smaller and shallower holes in the polymer layer are observed. Finally, the degradation of polymer thin layers can be explained by a synergistic effect of a local decrease of pH due to the hydrolysis of chlorpyrifos and the presence of acetonitrile which promotes the dissolution of deteriorated polymer layer.

## 5. Conclusions

In the context of developing surface acoustic wave transducers acting as passive, wireless cooperative targets for direct detection water table pollutant sensors, we have identified degradation processes when exposing PBMA, PIB and PECH to chlorpyrifos pesticide dissolved in water with concentration determined by the addition of acetonitrile. The degradation is explained following atomic force microscopy and Fourier-transform InfraRed characterization, and is the cause of unstable sensor response over timespans of several hours of exposure of the sensing area to the pesticide. Our work highlights the important role of deep investigations to determine the stability of polymers in their environmental conditions before they are used as a sensitive layer: the results presented here can be generalized to organophosphate direct detection by thin polymer coated sensors whose detection mechanism is more complex than surface absorption of molecule leading to mass loading as found in the literature. The development of new inert polymers for direct detection water table pollutant sensors is currently investigated.

## Figures and Tables

**Figure 1 sensors-22-01203-f001:**
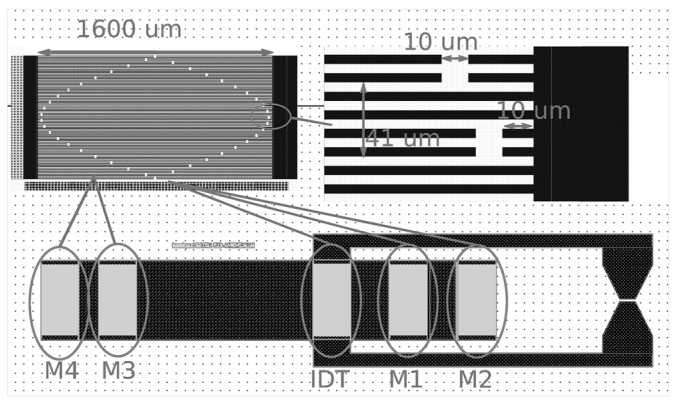
SAW sensor layout. The interdigitated transducer (IDT) is located between four mirrors (M1, M2, M3 and M4) positioned at different distances from the transducer in order to introduce different delays. The contact pads are used for the electrical connection of IDT. M1 acts for the reference measurement. The active areas between IDT and mirrors M1 or M3 and between mirrors M3 and M4 on the one hand, M1 and M2 on the other hand, are used for the deposition of the sensitive layer.

**Figure 2 sensors-22-01203-f002:**
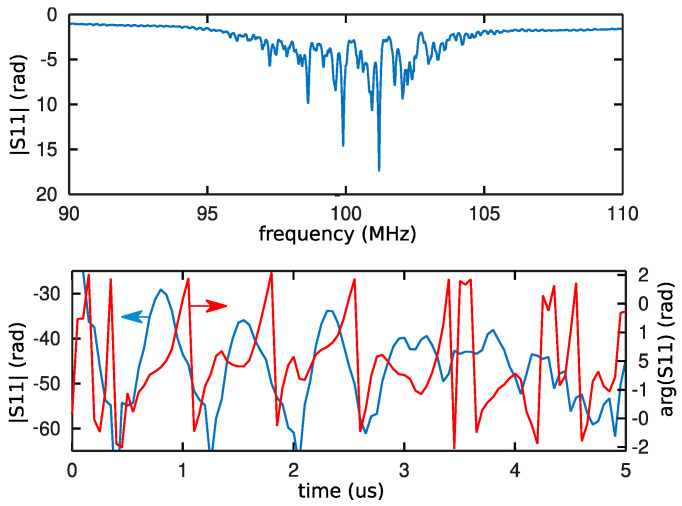
Frequency domain (**top**) and time domain (**bottom**) characterization of the reflective acoustic delay line used in the experiment depicted in this article. While the frequency domain transfer function indicates the spectral characteristics of the probed device, the four echoes as best viewed in the time domain magnitude (blue, bottom, at 0.8, 1.6, 2.4, and 3.2 μs). Fine acoustic velocity is measured through the phase variation (red) at the delay of maximum magnitude of each echo.

**Figure 3 sensors-22-01203-f003:**
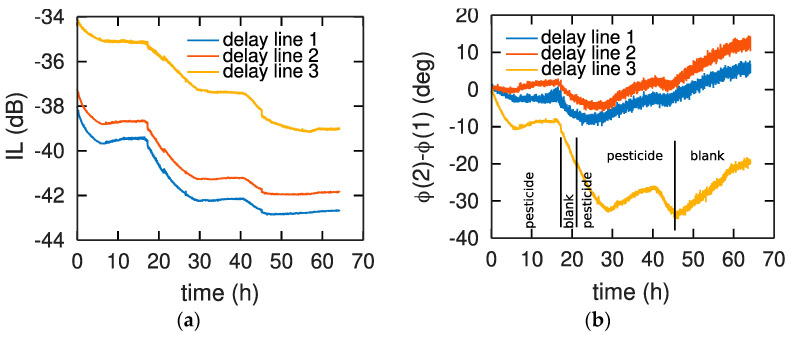
Evolution of (**a**) the insertion loss (IL) of echo 1 and (**b**) the phase difference between echo 1 and echo 2 separated by 800 ns. The measurement steps are indicated on the bottom for each chart since all steps were timestamped, with water-acetonitrile mixture (named blank) or water-acetonitrile containing chlorpyrifos solution (named pesticide) with PBMA as sensing layer.

**Figure 4 sensors-22-01203-f004:**
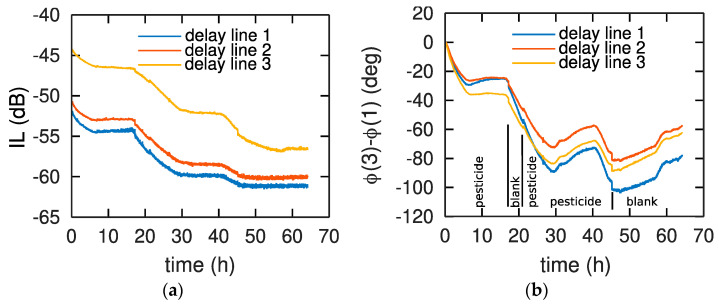
Evolution of (**a**) the insertion loss (IL) of echo 2 and (**b**) the phase difference between echo 2 and echo 3 separated by 1600 ns. The measurement steps are indicated on the bottom for each chart since all steps were timestamped, with water-acetonitrile mixture (named blank) or water-acetonitrile containing chlorpyrifos solution (named pesticide) with PBMA as sensing layer.

**Figure 5 sensors-22-01203-f005:**
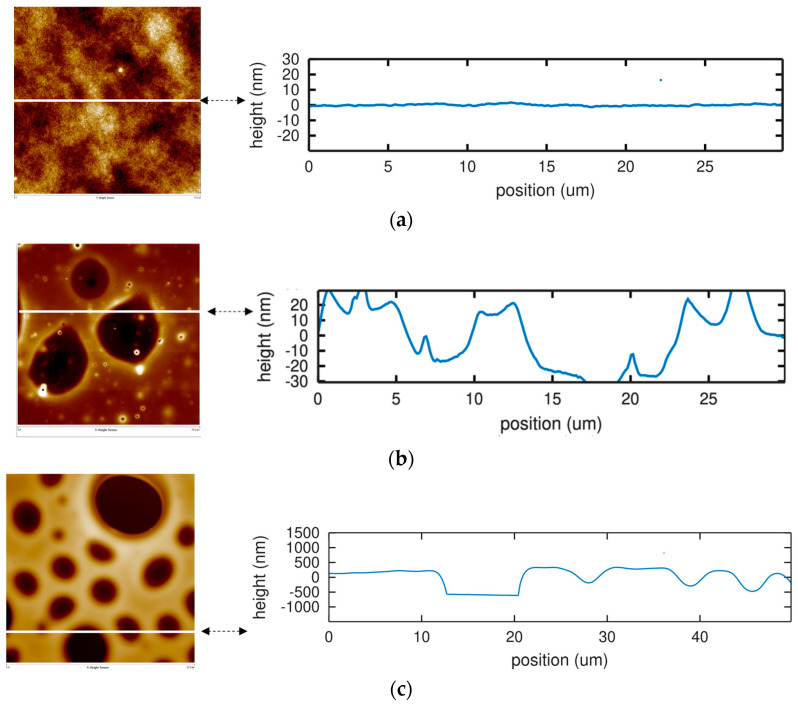
AFM topography scans (left) and corresponding height profile along the white line (right) of a silicon surface covered with a layer of PBMA (thickness: 500 nm). (**a**) Flat surface after deposition of PBMA by spin-coating. (**b**) The surface is still quite flat (RMS roughness: 11.3 nm) after immersion in water-acetonitrile (96:4) solution for 12 h despite some roughness increase with respect to (**a**). (**c**) After immersion in a water-acetonitrile solution of chlorpyrifos for 12 h then rinsed for 1 h with an acetonitrile-water solution (96:4), the surface of PBMA layer shows deep (200 nm) and crossing (500 nm) holes.

**Figure 6 sensors-22-01203-f006:**
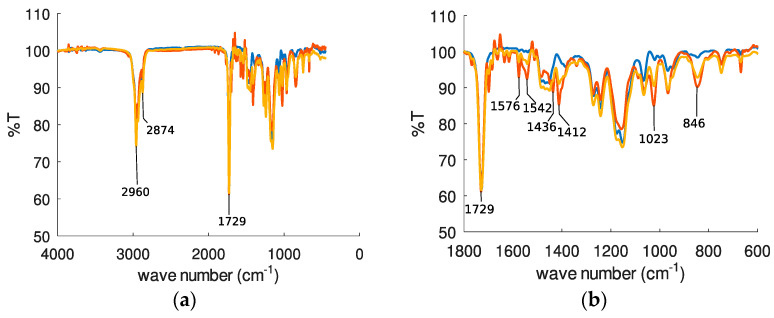
FT-IR spectra of a PBMA layer (thickness: 200 nm) deposited on a silicon substrate. In blue, spectrum of PBMA layer after deposition onto silicon substrate; in red, this layer immerged overnight in a water-acetonitrile solution containing 40 mg/L of chlorpyrifos; and in yellow, this layer washed with water-acetonitrile solution during 1.5 h (**a**) full extended spectrum; (**b**) zoom corresponding to 400–1800 cm^−1^ range of spectrum shown in (**a**).

**Figure 7 sensors-22-01203-f007:**
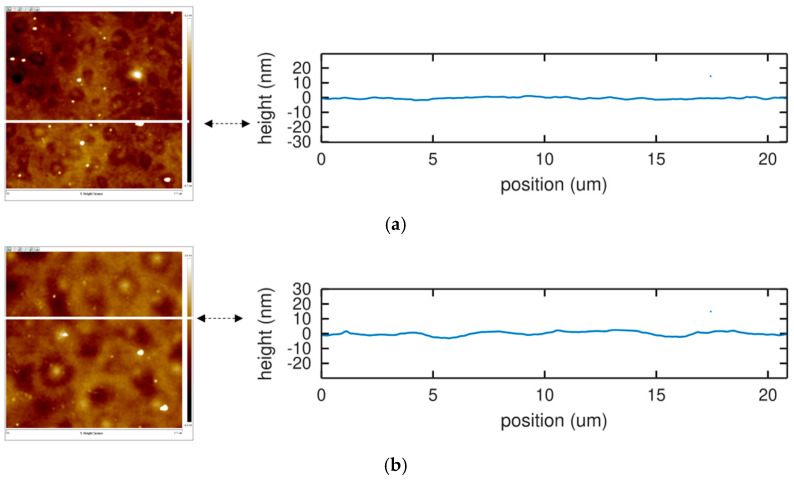
AFM topographies (left) and corresponding height profile along the white line (right) of a silicon surface covered with a spin-coated layer of PBMA (thickness: 500 nm). (**a**) Flat surface after immersion in aqueous PBS solution (pH = 6.2) for 12 h. (**b**) The surface roughness after immersion in aqueous PBS solution with chlorpyrifos solution for 12 h. The PBMA layer is not modified by the presence of chlorpyrifos.

**Figure 8 sensors-22-01203-f008:**
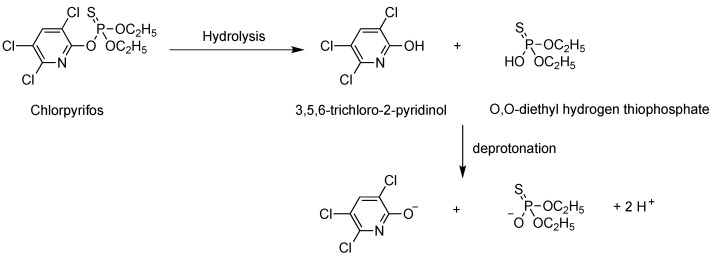
Hydrolysis of chlorpyrifos leading to 3,4,5-trichloro-2-pyrinol and O,O-diethyl hydrogen thiophosphate. These two molecules are weak acids which can lead to the formation of their corresponding base in water.

**Figure 9 sensors-22-01203-f009:**
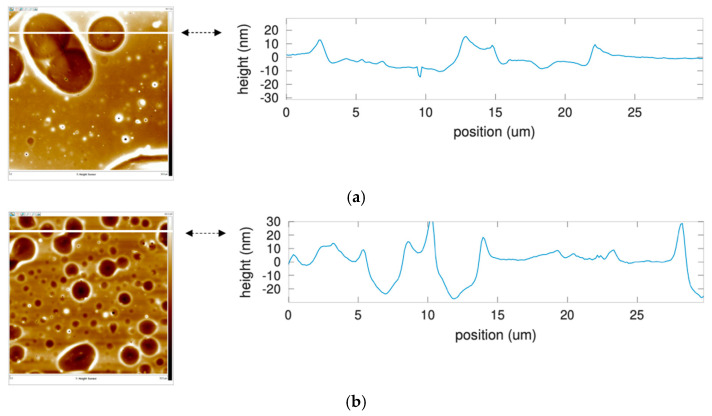
AFM topographies (left) and corresponding height profile along the white line (right) of a silicon surface covered with a spin-coated layer of PBMA (thickness: 500 nm) after immersion (**a**) in aqueous solution at pH = 5.0 for 12 h and (**b**) in water-acetonitrile (98:2) solution at pH = 5.0 for 12 h. The PBMA layer exhibits some holes with a depth of 20 nm and 50 nm in (**a**,**b**), respectively.

## Data Availability

The data are available upon reasonable request to the authors.

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
