# Peer review of "Degradation of Sub-Micrometer Sensitive Polymer Layers of Acoustic Sensors Exposed to Chlorpyrifos Water-Solution"

_sensors, 2022, doi:10.3390/s22031203_

Round 1

Reviewer 1 Report

This manuscript studies the degradation of polymer films on SAW sensors meant to detect chlorpyrifos.  The authors convincingly demonstrate physical and chemical changes in the polymer film that occur when exposed to unbuffered chlorpyrifos solution. It is argued that these changes are responsible for the irreproducible SAW measurements and that such degradation constitutes a strong limit for SAW chemical sensing.

This is useful work that should be published, after addressing a few concerning points:

The entire manuscript studies degradation when exposed to water-acetonitrile-chlorpyrifos. What is the basis for assuming 4% acetonitrile is negligible with PBMA, PIB, and PECH? It has not been convincingly demonstrated within this work that the results extend to water-chlorpyrifos.

Word choice: apparition should be appearance, exalted should be exhibited.

Figure 3 and 4 figures do not match captions. Fig 3 seems to be magnitude of echo 1 and phase distance between 1 and 2. Likewise Fig 4 seems to be magnitude of echo 2 and phase distance between echo 1 and 2. Furthermore Fig 3b and 4b are completely identical. The data alluded to in the figure captions would be more useful.

If acid generated by the hydrolysis of chlorpyrifos is the cause of degradation, it is straightforward to test both the pH of the solution and to reproduce that with a control experiment using e.g. citric acid-phosphate adjusted to the appropriate pH.

Author Response

This manuscript studies the degradation of polymer films on SAW sensors meant to detect chlorpyrifos.  The authors convincingly demonstrate physical and chemical changes in the polymer film that occur when exposed to unbuffered chlorpyrifos solution. It is argued that these changes are responsible for the irreproducible SAW measurements and that such degradation constitutes a strong limit for SAW chemical sensing.

This is useful work that should be published, after addressing a few concerning points:

The entire manuscript studies degradation when exposed to water-acetonitrile-chlorpyrifos. What is the basis for assuming 4% acetonitrile is negligible with PBMA, PIB, and PECH? It has not been convincingly demonstrated within this work that the results extend to water-chlorpyrifos.

We agree with this comment. We have added the description of the role of acetonitrile in the main manuscript (new lines 161-164). This solvent is used to increase the solubility of chlorpyrifos in a water solution. As described in Figure 5b, we carefully checked the role of acetonitrile by AFM in order to be sure that 4% acetonitrile has a negligible impact on a thin layer of the three investigated polymers.

Word choice: apparition should be appearance, exalted should be exhibited.

These errors have been corrected.

Figure 3 and 4 figures do not match captions. Fig 3 seems to be magnitude of echo 1 and phase distance between 1 and 2. Likewise Fig 4 seems to be magnitude of echo 2 and phase distance between echo 1 and 2. Furthermore Fig 3b and 4b are completely identical. The data alluded to in the figure captions would be more useful.

We apologize for these errors. We have updated Figure 4b. We have modified and clarified the captions and the corresponding main text, as suggested by the referee (new lines 180-194).

If acid generated by the hydrolysis of chlorpyrifos is the cause of degradation, it is straightforward to test both the pH of the solution and to reproduce that with a control experiment using e.g. citric acid-phosphate adjusted to the appropriate pH.

The relation between the pH at the solid-liquid nano-interface and the pH corresponding to the bulk solution is not straightforward. We have added the new reference 25 to support this statement in the text. Therefore, it is quite difficult to choose the value of pH to perform this experiment because we are not able to measure the local pH at the polymer-water interface.  In addition, it is impossible to investigate the stability of the polymers in a wide range of pH because the ionic force of the water solution will induce resistive shortcuts on IDTs, leading to the complete loss of response of the sensors.

Reviewer 2 Report

In general, the paper is good and is interesting, but it requires a major revision. There are some recommendations.

I recommend the authors to highlight the corrections and changes in the manuscript before the paper will undergo further assessment.

Therefore, the authors should revise the article by taking into consideration the following comments.

  • Authors should write an abstract according to the instructions of the subject journal, namely: The abstract should be a total of about 200 words maximum. The abstract should be a single paragraph and should follow the style of structured abstracts, but without headings: 1) Background: Place the question addressed in a broad context and highlight the purpose of the study; 2) Methods: Describe briefly the main methods or treatments applied. Include any relevant preregistration numbers, and species and strains of any animals used. 3) Results: Summarize the article's main findings; and 4) Conclusion: Indicate the main conclusions or interpretations. The abstract should be an objective representation of the article: it must not contain results which are not presented and substantiated in the main text and should not exaggerate the main conclusions.“
  • Authors should explain more about the novelty of their work in the introduction.
  • A lot of the introduction-related material is very forced, poorly articulated, and lacking a succinct structure. I would encourage the authors to consider consulting the article by George M. Whitesides, which was titled “Whitesides’ Group: Writing a Paper” such that the questions posed to the reader could be answered when reorganizing and redrafting a new introduction. Primarily, consider the following and outline your introduction (after rewriting your results and discussion sections) accordingly…
    • “Why did I do the work?”
    • “What were the central motivations and hypotheses?”
    • „What are the objectives of this work?“
    • “Why is the work important?”
    • “Who else has done what? How? What have we done previously?”
    • “What should the reader watch for in the paper?”
    • “What are the interesting high points?”
    • “What strategy did we use?”
    • “What should the reader expect as conclusion?”
  • In results and discussion, the authors should discuss on their results deeply. I strongly recommend expanding: Introduction, Conclusions and the Results sections. The aim should be to: 1) give a broader view of the literature on the topic and the current state-of-the-art; 2) clarify and discuss the novelty and the significance of the results obtained here, and compare them with those available in the literature, also including discussions on potential applications; 3) complete the manuscript with some additional, less basic results; 4) The authors should show the comparison between their results and previous works.
  • The authors must provide a greater discussion of the results.
  • The conclusions should be re-written, they should be specific and should not include as the initial paragraphs, which seems a summary, it must be enriched about discussion on solving problem.
  • The quality of the figures, for example Figure 1 in this document needs to be improved; the figures need to be larger in size so the data and labels can be clearly read.
  • Unify the axis description in Fig. 2 to Fig. 7.

Author Response

In general, the paper is good and is interesting, but it requires a major revision. There are some recommendations.

I recommend the authors to highlight the corrections and changes in the manuscript before the paper will undergo further assessment.

All corrections brought to the manuscript have been highlighted in yellow to be easily identified by the reviewers.

Therefore, the authors should revise the article by taking into consideration the following comments.

We appreciate the reference provided by the reviewer and copying the main items in their review. We have thoroughly assessed whether the recommendations we agreed with were addressed, and justify below those we do not agree with.

  • Authors should write an abstract according to the instructions of the subject journal, namely: The abstract should be a total of about 200 words maximum. The abstract should be a single paragraph and should follow the style of structured abstracts, but without headings: 1) Background: Place the question addressed in a broad context and highlight the purpose of the study; 2) Methods: Describe briefly the main methods or treatments applied. Include any relevant preregistration numbers, and species and strains of any animals used. 3) Results: Summarize the article's main findings; and 4) Conclusion: Indicate the main conclusions or interpretations. The abstract should be an objective representation of the article: it must not contain results which are not presented and substantiated in the main text and should not exaggerate the main conclusions.“

The abstract is 80-word long formatted as a single paragraph and matches the MDPI template as does the whole manuscript. Background: "The detection of organophosphates, a wide class of pesticides, in water-solution has a huge impact in environmental monitoring." Methods: "Acoustic transducers are used to design passive wireless sensors for the direct detection of pesticides in water-solution by using tailored polymers as sensitive layers." Results: "We demonstrate by combining analytical chemistry tools that organophosphate molecules strongly alter polymer layers widely-used in acoustic sensors in presence of water." Hence, unless the reviewer has some specific recommendation, we fail to identify how our current abstract does not match their outline.

  • Authors should explain more about the novelty of their work in the introduction.

The introduction ends with "In this work, we monitor the acoustic response of SAW devices functionalized with PECH, PIB and PBMA thin films exposed to chlorpyrifos water-solution in environmental conditions. Then we assess the instability of the measurements by considering analytical characterization of the thin films prior and after exposure to chlorpyrifos using Fourier-Transform InfraRed Spectroscopy (FT-IR) and Atomic Force Microscopy (AFM). We demonstrate the irreversible degradation of the thin polymer films exposed to chlorpyrifos in water-solution. This instability of sensitive layers seems to be a strong limitation to the popularization of this kind of acoustic sensors in environmental challenge and water resource management in practical deployment scenarios." which summarizes the original contribution of this work, especially in the context of past investigations claiming to use these polymers for pesticide detection in media irrelevant of practical deployment. Hence, unless the reviewer has some specific recommendation, we fail to identify how our current introduction does not match their outline.

  • A lot of the introduction-related material is very forced, poorly articulated, and lacking a succinct structure. I would encourage the authors to consider consulting the article by George M. Whitesides, which was titled “Whitesides’ Group: Writing a Paper” such that the questions posed to the reader could be answered when reorganizing and redrafting a new introduction. Primarily, consider the following and outline your introduction (after rewriting your results and discussion sections) accordingly…
    • “Why did I do the work?”
    • “What were the central motivations and hypotheses?”
    • „What are the objectives of this work?“
    • “Why is the work important?”
    • “Who else has done what? How? What have we done previously?”
    • “What should the reader watch for in the paper?”
    • “What are the interesting high points?”
    • “What strategy did we use?”
    • “What should the reader expect as conclusion?”

We have followed the classical outline of chemical sensing articles with an introduction summarizing our knowledge of the state of the art, context of the investigation and citing prior work. Follows a Materials and Methods summarizing the experimental procedure and measurement protocols. Follows a Results section summarizing the main observations from the various measurements’ techniques. Follows a Discussion section analyzing the results and providing some possible explanation. All these sections are summarized in the Conclusion which, as opposed to following statement by the reviewer, should be in our opinion a summary of the main results motivating the reader to address the full manuscript after browsing through the abstract for the motivation of the work and the conclusion to grasp the highlights of the work. Hence, unless the reviewer has some specific recommendation, we fail to identify how our current manuscript layout does not match their outline.

  • In results and discussion, the authors should discuss on their results deeply. I strongly recommend expanding: Introduction, Conclusions and the Results sections. The aim should be to: 1) give a broader view of the literature on the topic and the current state-of-the-art;

We respectfully disagree with this statement: we collect the literature review in the introduction providing the context of the work and past results outlining the experimental scheme developed in the current work, and we believe the Results section should only be filled with novel results from our measurements and not refer to prior work from other groups.

  • 2) clarify and discuss the novelty and the significance of the results obtained here, and compare them with those available in the literature, also including discussions on potential applications; 3) complete the manuscript with some additional, less basic results; 4) The authors should show the comparison between their results and previous works.

This manuscript does not involve a practical application but demonstrates how some of the proposed functionalization methods are NOT compatible with practical applications, as stated in "This instability of sensitive layers seems to be a strong limitation to the popularization of this kind of acoustic sensors in environmental challenge and water resource management in practical deployment scenarios." We fail to understand how providing the exact same review as those provided in https://www.mdpi.com/2077-0375/11/10/751/review_report and https://www.mdpi.com/1099-4300/22/7/771/review_report can help this particular manuscript.

  • The authors must provide a greater discussion of the results.

We have analyzed and referred to all curves provided in the Results section. Hence, unless the reviewer has some specific recommendation, we fail to identify how our current discussion section does not match their objective

  • The conclusions should be re-written, they should be specific and should not include as the initial paragraphs, which seems a summary, it must be enriched about discussion on solving problem.

As stated earlier, we respectfully disagree with this use of the conclusion and fail to understand how the current version can be improved, unless the reviewer has some specific recommendation.

  • The quality of the figures, for example Figure 1 in this document needs to be improved; the figures need to be larger in size so the data and labels can be clearly read.

Indeed Figs. 3 and 4 had been erroneously duplicated and did not match their caption. These figures have been updated and the captions corrected accordingly (as also requested by reviewer 1).

  • Unify the axis description in Fig. 2 to Fig. 7.

We fail to understand this statement as those Figures cover widely different topics and illustrate very different measurement techniques. Hence, only Figs 3 and 4 have been updated. The reviewer might want to notice that all AFM cross sections have been displayed with a common height range for easy comparison by the reader, except for the strongly degraded polymer layers with the holes going through the thin film in which the Y-scale has been obviously magnified.

Round 2

Reviewer 1 Report

The entire manuscript studies degradation when exposed to water-acetonitrile-chlorpyrifos. What is the basis for assuming 4% acetonitrile is negligible with PBMA, PIB, and PECH? It has not been convincingly demonstrated within this work that the results extend to water-chlorpyrifos.

We agree with this comment. We have added the description of the role of acetonitrile in the main manuscript (new lines 161-164). This solvent is used to increase the solubility of chlorpyrifos in a water solution. As described in Figure 5b, we carefully checked the role of acetonitrile by AFM in order to be sure that 4% acetonitrile has a negligible impact on a thin layer of the three investigated polymers.

Although the authors have improved the manuscript with these changes, their response reinforces my original objection: just as the acetonitrile assists in the dissolution of chlorpyrifos in water, there may be a synergistic effect in which acetonitrile increases the infiltration of chlorpyrifos in the sensor films, possibly leading to increased degradation. Although I have so far taken the claim that Figure 5B shows negligible degradation at face value for SAW sensors, many other sensors (QCM, SPR, ellipsometry) would be measurably affected by such a change in roughness that is clearly caused by 4% acetonitrile. 

A control experiment that shows the behavior of water-chlorpyrifos would be ideal, however if that is impossible due to solubility, a factorial design of experiments varying of water-acetonitrile-chlorpyrifos (W:A:C, mL:mL:mg) could be acceptable. Specifically, 980:20:40, 960:40:20, and 980:20:20 when combined with your existing 960:40:40 experiment would be able to detect the individual effects of acetonitrile and chlorpyrifos as well as their interaction.

If acid generated by the hydrolysis of chlorpyrifos is the cause of degradation, it is straightforward to test both the pH of the solution and to reproduce that with a control experiment using e.g. citric acid-phosphate adjusted to the appropriate pH.

The relation between the pH at the solid-liquid nano-interface and the pH corresponding to the bulk solution is not straightforward. We have added the new reference 25 to support this statement in the text. Therefore, it is quite difficult to choose the value of pH to perform this experiment because we are not able to measure the local pH at the polymer-water interface. 

Ref. 25 is a simulation paper regarding hydrogen generating electrodes. According to the simulations of ref. 25, unless your IDT is generating a stronger potential than -0.5 V, the reference is completely irrelevant and should be removed/replaced. Assuming the potential I would not expect chlorpyrifos to have any altered rate of hydrolysis at the polymer sensor interface but could be convinced by a reference. At any rate I am interested in the correlation with the bulk pH and buffer strength regardless of the specific local pH gradients.

In addition, it is impossible to investigate the stability of the polymers in a wide range of pH because the ionic force of the water solution will induce resistive shortcuts on IDTs, leading to the complete loss of response of the sensors.

You do not need to perform the control pH experiment on the sensors, a simple film with AFM height map data would suffice.

Author Response

We have appreciated to read your enthusiastic and constructive comments  about our work. As you have suggested, we have seriously considered your different comments and suggestions . We have made extensive modifications and improvement in the revised manuscript that address the different points raised by the referee in the report. All significant changes in the manuscript have been highlighted in yellow in the revised version for your convenience, including multiple figures resulting from additional experiments.

Although the authors have improved the manuscript with these changes, their response reinforces my original objection: just as the acetonitrile assists in the dissolution of chlorpyrifos in water, there may be a synergistic effect in which acetonitrile increases the infiltration of chlorpyrifos in the sensor films, possibly leading to increased degradation. Although I have so far taken the claim that Figure 5B shows negligible degradation at face value for SAW sensors, many other sensors (QCM, SPR, ellipsometry) would be measurably affected by such a change in roughness that is clearly caused by 4% acetonitrile. 

We agree that the impact of acetonitrile would be detectable using direct detections sensors (QCM, etc.) but this impact remains smaller than the global impact of exposing to chlorpyrifos in a water-acetonitrile solution, as shown by the RMS roughness values. The text indeed mentions “that the layers are slightly affected” (Lines 249-250). However, the synergistic effect suggested by the reviewer has been investigated as described below.

A control experiment that shows the behavior of water-chlorpyrifos would be ideal, however if that is impossible due to solubility, a factorial design of experiments varying of water-acetonitrile-chlorpyrifos (W:A:C, mL:mL:mg) could be acceptable. Specifically, 980:20:40, 960:40:20, and 980:20:20 when combined with your existing 960:40:40 experiment would be able to detect the individual effects of acetonitrile and chlorpyrifos as well as their interaction.

We have performed many additional experiments to highlight the impact of acid as advised by the reviewer. The main text now includes Figure 9 demonstrating the synergetic impact of acid and chlorpyrifos in degrading the polymer thin film. Additionally, in supplementary materials, the new Figure S7, demonstrates the effect of the concentration of acetonitrile on the thin film degradation at a fixed concentration on chlorpyrifos, as also suggested by reviewer. Our conclusions are thus confirmed as stated in the main text in page 9:

“As the chlorpyrifos is hydrophobic, this compound is poorly soluble in the water-acetonitrile (96:4) solution. Therefore, chlorpyrifos can be mainly adsorbed onto the PBMA thin layer, because PBMA contains alkyl chains promoting van der Waals interaction with the aromatic core of chlorpyrifos. Finally, as chlorpyrifos is mainly located at the water-polymer interface, the formation of H+, due to the hydrolysis of chlorpyrifos, is enhanced at this interface, which leads to the enhancement of acidic degradation of the PBMA layer as observed in AFM images. In addition, pH can be very different at solid-liquid interface from the solution as it has been reported in the case of polymer-water interface [25]. Figure 9 emphasizes the role of acetonitrile in dissolving the products coming from the acidic hydrolysis of PBMA polymer strands. Indeed, exposition of the PBMA layer to the same pH without acetonitrile leads to surface degradation but shallower holes (a peak-to-valley depth of 20 nm with RMS roughness of 4.8 nm, Figure 9a; a peak-to-valley depth of 40 nm with RMS roughness of 10.3 nm, Figure 9b). Similarly, Figure S7 confirms this conclusion by replacing constant pH solution with constant chlorphyrifos concentration and varying acetonitrile concentration: halving the latter significantly reduces the degraded polymer solubilization capability and hence smaller and shallower holes in the polymer layer are observed. Finally, the degradation of polymer thin layers can be explained by a synergistic effect of a local decrease of pH due to the hydrolysis of chlorpyrifos and the presence of acetonitrile which promotes the dissolution of deteriorated polymer layer.

Ref. 25 is a simulation paper regarding hydrogen generating electrodes. According to the simulations of ref. 25, unless your IDT is generating a stronger potential than -0.5 V, the reference is completely irrelevant and should be removed/replaced. Assuming the potential I would not expect chlorpyrifos to have any altered rate of hydrolysis at the polymer sensor interface but could be convinced by a reference. At any rate I am interested in the correlation with the bulk pH and buffer strength regardless of the specific local pH gradients.

We agree, we have replaced the ref 25 by a new one, entitled: “Evaluation of Interfacial pH Using Surface Forces Apparatus Fluorescence Spectroscopy”. Our statement about chlorpyrifos hydrolysis at the polymer interface is more detailed in the main text by this new sentence: “Such a degradation effect is enhanced at the polymer interface considering the poor solubility of the organophosphate molecule in water as indicated by the need for adding acetonitrile.”

In addition, it is impossible to investigate the stability of the polymers in a wide range of pH because the ionic force of the water solution will induce resistive shortcuts on IDTs, leading to the complete loss of response of the sensors.

You do not need to perform the control pH experiment on the sensors, a simple film with AFM height map data would suffice.

This point has been addressed above.